# Advances in Imaging for Assessing Pelvic Endometriosis

**DOI:** 10.3390/diagnostics12122960

**Published:** 2022-11-26

**Authors:** Stefano Guerriero, Silvia Ajossa, Mariachiara Pagliuca, Antonietta Borzacchelli, Fabio Deiala, Serena Springer, Monica Pilloni, Valeria Taccori, Maria Angela Pascual, Betlem Graupera, Luca Saba, Juan Luis Alcazar

**Affiliations:** 1Centro Integrato di Procreazione Medicalmente Assistita (PMA) e Diagnostica Ostetrico-Ginecologica, Azienda Ospedaliero Universitaria-Policlinico Duilio Casula, 09042 Monserrato, Italy; 2Department of Obstetrics and Gynecology, University of Cagliari, 09042 Monserrato, Italy; 3Department of Obstetrics and Gynecology, Policlinico Universitario Duilio Casula, University of Cagliari, 09042 Monserrato, Italy; 4Surgical and Health Sciences, Department of Medical, University of Trieste, 34127 Trieste, Italy; 5Department of Obstetrics, Gynecology and Reproduction, Hospital Universitari Dexeus, 08028 Barcelona, Spain; 6Department of Radiology, Azienda Ospedaliero Universitaria-Policlinico Duilio Casula, Monserrato, 09042 Cagliari, Italy; 7Department of Obstetrics and Gynecology, School of Medicine, University of Navarra, 31009 Pamplona, Spain

**Keywords:** endometriosis, transvaginal ultrasound, magnetic resonance imaging

## Abstract

In recent years, due to the development of standardized diagnostic protocols associated with an improvement in the associated technology, the diagnosis of pelvic endometriosis using imaging is becoming a reality. In particular, transvaginal ultrasound and magnetic resonance are today the two imaging techniques that can accurately identify the majority of the phenotypes of endometriosis. This review focuses not only on these most common imaging modalities but also on some additional radiological techniques that were proposed for rectosigmoid colon endometriosis, such as double-contrast barium enema, rectal endoscopic ultrasonography, multidetector computed tomography enema, computed tomography colonography and positron emission tomography–computed tomography with 16α-[18F]fluoro-17β-estradiol.

## 1. Introduction

Endometriosis is a common chronic disease affecting 5–10% of women of reproductive age globally [1]. Histologically, this disease is defined by the presence of endometrium-like tissue (epithelium and/or stroma) outside the endometrium and myometrium, usually with an associated inflammatory process [1] and several different clinical presentations that are mainly based on the presence of pain but also infertility [2]. According to the International Working Group of the AAGL (American Association of Gynecologic Laparoscopists), ESGE (European Society of Gynecologic Endoscopy), ESHRE (European Society of Human Reproduction and Embryology) and WES (World Endometriosis Society), superficial endometriosis is defined by the presence of endometrium-like tissue involving the peritoneal surface. The lesions can have different appearances and colors e.g., clear and black [2]. In contrast, the same group defines deep endometriosis (DE) as endometrium-like tissue lesions in the abdomen that extend on or under the peritoneal surface. They are usually nodular, able to invade adjacent structures, and associated with fibrosis and the disruption of normal anatomy [2]. Ovarian endometriosis is defined as endometrium-like tissue in the form of ovarian cysts [2]. Endometriomas may be either invagination cysts or true cysts, with the cyst wall also containing endometrial-like tissue and dark blood-stained fluid [2].

The ovary is a common site affected by endometriosis [3]. The intra-pelvic localizations of endometriosis include involvement of the uterosacral ligaments (USLs), vagina, rectum, rectovaginal space, or bladder (frequently at the posterior bladder dome) [4]. Endometriosis can also affect extra-pelvic organs, such as the abdominal wall, the diaphragm and nerves (for example sciatic nerves) [5], but these topics were covered in a previous review [6].

In recent years, due to the development of standardized diagnostic protocols associated with an improvement in technology, the diagnosis of pelvic endometriosis via imaging is becoming a reality. In particular, transvaginal ultrasound (TVUS) and magnetic resonance imaging (MRI) are today the two imaging techniques that can accurately identify the majority of the previously described phenotypes of endometriosis. This review focuses not only on these most common imaging modalities but also on some additional radiological techniques that were suggested for rectosigmoid colon endometriosis and other specific purposes, such as double-contrast barium enema, rectal endoscopic ultrasonography, multidetector computed tomography enema, computed tomography colonography and positron emission tomography–computed tomography with 16α-[18F]fluoro-17β-estradiol.

## 2. Transvaginal Ultrasound

TVUS is surely the first-line technique for the diagnosis of endometriosis [7]. The possibility to perform a dynamic ultrasonographic examination can provide additional information that is not easily obtained with other imaging modalities [7].

### 2.1. Ovarian Endometriosis

The ultrasound features of ovarian endometriosis vary in pre- and postmenopausal patients [8,9]. The most common ultrasound appearance of ovarian endometriosis is the typical “unilocular cyst with ground glass echogenicity of the cyst fluid” [9]. This appearance became less common with increasing age, in particular after 35 years old [8]. Although the mean diameter remains the same in the different ages of premenopausal women, endometriomas with papillations are more common in older women, with 3% in younger women compared with 14% of women ≥ 45 years old [8]. In a Cochrane review, Nisenblat et al. [10] found that the sensitivity and specificity for a diagnosis of endometrioma using TVUS are 93% and 96%, respectively. The age of the patients seems to also interfere with the diagnostic accuracy [8]. As a matter of fact, subjective impressions of the presence of endometrioma showed a sensitivity of around 90% in younger patients that decrease to 70% after 45 years of age. In contrast, the specificity remains the same (97–98%).

### 2.2. Superficial Endometriosis

Although some authors suggest that some findings (presence of thickened pericolic fat) correlate with the presence of superficial endometriosis, no reproducibility study has been published [11]. Robinson et al. [12] found a poor sensitivity (51%) and specificity (55%) of transvaginal ultrasound markers in the diagnosis of superficial endometriosis of the USL. Reid et al. found that, in the absence of ovarian endometriosis and DE nodules, ovarian immobility was associated with superficial endometriosis of the pelvic side wall [13]. Leonardi et al. described a novel technique, the so-called “sonoPODgraphy”, that apparently allows for better detection of superficial endometriosis [14].

### 2.3. Deep Endometriosis

Regarding DE, the consensus from the International Deep Endometriosis Analysis (IDEA) group [15] proposed four basic sonographic steps when examining women with suspected or known endometriosis: evaluation of the uterus and adnexa (signs of adenomyosis and endometriomas), evaluation of “soft markers”, assessment of the status of the pouch of Douglas (POD) using the “sliding sign”, and assessment of DE nodules in the anterior and posterior compartments (bladder, vaginal vault, USLs, bowel, rectum, rectosigmoid junction and sigmoid colon).

The identification of “soft markers” is an easy-to-learn assessment that can even be performed by less experienced operators [16]. “Soft markers” were designed to identify patients who would most likely benefit from a more detailed scan by an expert examiner [16]. Markers that showed a correlation with the presence of DE are the absence of the “sliding sign” (applying gentle pressure to the cervix to mobilize the uterus to determine whether the anterior rectum glides freely or not over the posterior vagina, cervix and uterus) and/or the presence of “kissing ovaries” (both ovaries located in close proximity or touching each other in the POD) in patients with clinical suspicion of endometriosis [16]. In a recent systematic review and meta-analysis, Alcazar et al. [17] found that the “sliding sign” has good diagnostic performance for predicting POD obliteration and bowel involvement in women with suspected endometriosis. In addition, ovarian immobility in TVUS is significantly associated with ipsilateral pelvic pain, USL and pelvic sidewall superficial endometriosis, endometrioma, posterior compartment DE and POD obliteration [13]. Another very rare localization is the presence of tubal endometriosis. The only study present in the literature [18] found that hydrosalpinx as an ultrasonographic sign used alone is characterized by a high specificity but low sensitivity for the detection of tubal endometriosis, which can be improved by the addition of other ultrasonographic markers previously described.

According to the IDEA consensus [15], the location of DE nodule localizations should be divided into two different pelvic compartments: the anterior and the posterior. The anterior compartment includes the urinary bladder, uterovesical region and ureters, while the posterior compartment includes the posterior vaginal fornix, anterior rectum/anterior rectosigmoid junction, retrocervical space and sigmoid colon [15].

Recent studies evaluated the prevalence, diagnosis and clinical features of DE involving the parametrium, considering it as a third compartment in addition to the anterior and posterior ones. According to the #Enzian Classification for endometriosis [19], the mediolateral/dorsolateral compartments include the USL, cardinal ligament and pelvic sidewall. From the anatomical point of view, the lateral compartment includes the parametrium, paracolpium, pelvic floor, the parametrial segment of the ureters, nerves (inferior hypogastric plexus and obturatory nerve), ovarian fossae and pelvic sidewalls [20,21]. Generally, TVUS should be performed with a scarcely filled bladder. However, if bladder endometriosis is suspected, patients should be asked not to empty their bladder before the ultrasound examination [15]. The image orientation (up or down) depends on the choice of the operator [22].

#### 2.3.1. Anterior Endometriosis

The DE in the anterior compartment can include hypoechoic linear or spherical lesions, with or without regular contours, cystic spaces, hyperechoic foci and/or regular contours involving the muscularis (most commonly), (sub)mucosa of the bladder (lesions that involve only the serosa represent superficial endometriosis) or the uterovesical space [15] (Figure 1). Albeit not formally assessed in prospective studies, the obliteration of the uterovesical space can be evaluated using the sliding of these structures. If the posterior bladder slides freely over the anterior uterine wall, the uterovesical space is considered non-obliterated. On the other hand, if the bladder does not slide freely, the uterovesical space can be considered obliterated [15]. In a meta-analysis, Guerriero et al. [23] found that for the detection of bladder endometriosis, the sensitivity and specificity were 62% and 100%, respectively. Similar results were reported in a more recent meta-analysis [24].

In the rectovaginal space (RVS) the DE nodules are described as lesions below a horizontal plane that passes along the lower margin of the posterior lip of the cervix under the peritoneum [15]. (Figure 2). Guerriero et al. [23] found that in the detection of endometriosis in the RVS, the sensitivity and specificity were 49% and 98%, respectively.

Experienced sonographers can reliably identify silent ureteral involvement due to endometriosis with TVUS [23]. The most common location of ureteral stenosis is 3–4 cm above the vesicoureteral junction and, in about half of the cases, ureteral dilatation involvement is not associated with hydronephrosis on abdominal ultrasound [25]. In a prospective observational study that enrolled 848 patients with chronic pelvic pain, Pateman et al. [26] found that the diagnosis of ureteral endometriosis had a sensitivity and specificity of 92% and 100%, respectively. Although a key point for surgical planning includes the close relationship with the ureteral path [27], the measurement of the distance between the nodule of the ovarian fossae and the ureteral path is not described in the literature.

#### 2.3.2. Posterior and Lateral Endometriosis

As for the USL, the lesions appear as nodules with regular or irregular margins and often with hyperechoic points, or as a linear hypoechoic thickening with regular or irregular margins [15] (Figure 3). In the detection of USL endometriosis, the sensitivity and specificity are 53% and 93%, respectively [23]. Vaginal lesions should be suspected when the posterior vaginal fornix is thickened with or without surrounding cystic anechoic areas or if there is a hypoechoic nodule, homogeneous or inhomogeneous, with or without large cystic areas [23]. (Figure 4). In the detection of vaginal endometriosis, the sensitivity and specificity are 58% and 96%, respectively [23]. A recent meta-analysis reported similar data with a sensitivity of 52% and a specificity of 98% [24].

A nodule on the rectosigmoid colon can appear as an irregular hypoechoic nodule in the anterior wall of the rectosigmoid colon with or without hypoechoic or, rarely, a hyperechoic focus with visible retraction and adhesions [15] (Figure 5). Sensitivity and specificity in the detection of endometriotic lesions in the rectosigmoid colon are 91% and 97%, respectively [28]. In addition, TVUS enables the accurate assessment of a rectosigmoid DE lesion size with moderate-to-good reliability and correlation for lesion thickness measurements, which may be essential for diagnosis, surgical risk assessment and planning of surgical treatment [29]. TVUS is also reliable at measuring the lesion-to-anal-verge distance of rectosigmoid DE, thereby estimating the height of the bowel anastomosis in patients undergoing a-full-thickness resection either by discoid resections or segmental resections [30,31]. A recent prospective study performed by the IDEA group found a higher TVUS detection rate of DE overall than that reported by the most recent meta-analysis on the topic (sensitivity of 79%), but with a lower specificity [32].

According to the study of Exacoustos et al. [33], parametrial involvement is suggested by infiltrating, irregular and hypoechogenic tissue extending laterally to the cervix or vagina. Parametrial endometriosis involvement occurs more frequently in patients with severe endometriosis, possibly with hypogastric/sacral plexus or sciatic nerve involvement [34,35]. In a recent meta-analysis, Guerriero et al. [36] found that in the detection of endometriosis in the parametrium, the sensitivity and specificity are 31% and 98%, respectively. This low sensitivity can be explained by the fact that specific training is needed to properly visualize this anatomical area [36]. Assessing the USL might suggest a thorough evaluation of the parametrium [37].

#### 2.3.3. Additional Ultrasonographic Techniques and TVUS Classification Systems

To further increase the diagnostic accuracy of TVUS, there are some additional techniques that can be performed. The simplest additional technique is the tenderness approach [38]. It is a modality of TVUS that is performed by increasing the amount of ultrasound gel inside the probe cover by introducing 12 mL of ultrasound gel instead of the usual 4 mL. The gel lid creates a stand-off that allows for the visualization of adjacent tissues to the probe. It is also useful to evaluate painful sites evocated by the gentle pressure of the probe, even without abdominal palpation. No data are present on the level of expertise needed. This technique known as “tenderness-guided” TVUS [39] shows high specificity and sensitivity in the detection of vaginal and rectovaginal space endometriosis. Good specificity associated with a lower sensitivity was obtained in the diagnosis of DE of USL, rectosigmoid involvement or anterior DE [38]. This technique was meant to reproduce physical examination that shows sensitivities in endometriosis diagnosis of 41% for the ovary, 50% for the USL, 76% for the POD, 73% for the vagina, 78% for the RVS, 25% for the urinary bladder and 39% for the rectosigmoid [40].

Regarding the need for bowel preparation before a TVUS examination, Ros et al. [41] concluded that it was well-tolerated and useful to aid in the detection and more precise description of rectosigmoid nodules. Ferrero et al. [42], on the other hand, found that bowel preparation does not improve the diagnostic performance of TVUS in detecting rectosigmoid endometriosis and in assessing the characteristics of endometriotic nodules.

In addition, office gel sonovaginography (SVG) appears to be an effective outpatient imaging technique for the prediction of bowel DE, with higher accuracy for the prediction of rectosigmoid compared with anterior rectal DE [43]. SVG can improve the detection of DE involving the rectovaginal space by creating an acoustic window between the transvaginal probe and the surrounding vaginal structures [44]. A further technique that is used is rectal water-contrast transvaginal ultrasonography (RWC-TVUS). Although RWC-TVUS and SVG have similar accuracies in the diagnosis of DE, RWC-TVUS has a higher performance when assessing the characteristics of rectosigmoid endometriosis. Like conventional TVUS, a limitation of enhanced TVUS techniques is that they cannot diagnose endometriotic nodules located above the rectosigmoid, as they are beyond the field of view of ultrasonography [45]. Additional evaluation through the abdominal wall using a high-resolution linear transducer might be needed in some cases, for example, to assess cecal or appendicular endometriosis. Some authors [46] suggested using a laparoscopic intraoperative 12.4 MHz ultrasonographic transducer placed on the surface of the rectosigmoid nodules to study the appearances and to perform measurements. Due to the few cases evaluated, it is probably too early to introduce this practice in daily management.

Based on the high diagnostic accuracy reported in the literature, of all classification and staging systems for endometriosis, many are based on ultrasound. Table 1 summarizes the most recent (of the last 10 years) endometriosis classification/staging system based on ultrasound.

In order to achieve the quality of care of a tertiary center obtained by expert examination in regular practices, in particular in the USA, Young et al. [53] suggested accessing the endometriotic specific sites and training sonographers, as well as radiologic and gynecologic sonologists, in the sonographic features of deep endometriosis and other associated signs. In addition, Deslandes et al. [54] suggested that doubling the scan time should be considered due to a demonstrated additional average time of 5.4 min in endometriosis sonographic assessment in comparison with routine transvaginal ultrasound.

Three-dimensional TVUS (3D TVUS) can also be used for the evaluation of DE. This technique allows for the possibility of obtaining volumes that can be reconstructed manyfold after a single sweep with an infinite number of viewing planes [55]. In particular, it is possible to obtain the coronal plane of the lesion, which is a plane that is practically impossible to obtain with the use of 2D TVUS [56]. In addition, 3D TVUS rendering can allow for a better analysis of a nodule because 3D reconstruction might make the irregular shapes and borders more evident [57], as well as the use of volume contrast imaging (VCI) with 2–4 mm slices. The possibility to re-evaluate stored 3D volumes, the so-called “virtual navigation”, can be used as an educational tool to improve the learning curve of less expert operators. In fact, Guerriero et al. [57] found that the combined use of real-time TVUS and offline 3D volume virtual navigation was helpful to improve training in a short time (2 weeks) for the ultrasound assessment of DE. Regarding the diagnoses superficial endometriosis but also endometriosis of ureters and parametrium, there are presently no studies in the literature that describe the accuracy of 3D TVUS. Barra et al. [58] found that 3D reconstructions do not improve the performance of 2D TVUS in diagnosing the presence and characteristics of bladder endometriosis. However, the use of virtual organ computer-aided analysis (VOCAL) allows for a significantly better volume assessment. Guerriero et al. [59] found that for 3D TVUS, the sensitivity and specificity in the diagnosis of rectosigmoid colon endometriosis were 91% and 97%, respectively, similar to 2D, while in the diagnosis of endometriosis of the USL, rectovaginal space and vaginal fornix overall, the sensitivity and specificity, thanks to the available coronal plane, are 87% and 94%, respectively. In another study, retrocervical and rectosigmoid endometriotic nodules displayed significantly different three-dimensional sonographic mean gray values, which allowed for a quantitative evaluation with extremely good reproducibility [60]. In addition, introital 3D ultrasonography seems to be an effective method for the diagnosis of endometriosis of the rectovaginal septum [61]. In a recent study, Guerriero et al. [62] found that the typical ultrasonographic sign of rectosigmoid endometriosis is reasonably recognizable to observers with different levels of expertise when assessed in stored 3D volumes.

## 3. Magnetic Resonance Imaging (MRI)

MRI is commonly used for the diagnosis of endometriosis. However, a consensus on the best descriptors of DE on MRI is lacking. The European Society of Urogenital Radiology (ESUR) [63] guidelines recommend MRI as a second-line examination technique after TVUS in the evaluation of endometriosis. In addition, ESUR recommends MRI for preoperative staging before surgery for predicting the diagnosis of multiple sites of DE in the case of equivocal TVUS or in a symptomatic patient with negative TVUS findings [63]. Additionally, the Society of Abdominal Radiology (SAR) Disease Focused Panel on Endometriosis proposed a new MRI lexicon for describing findings on MRI [64].

In order to obtain optimal visualization of the anterior compartment, it is suggested to perform the MR exam with moderate repletion of the patient’s bladder; this is requested because an overfilled bladder could determine the detrusor contractions with subsequent obliteration of the adjacent recesses, thus compromising the identification of small parietal nodules [63,65]. While for ESUR, fasting and bowel preparation are recommended before MRI in the evaluation of DE, while vaginal opacification is considered an option [63], though it is not necessary for the adequate assessment of endometriotic lesions.

It is possible to perform the MR exam with different magnetic field strengths (1.5 T or 3 T). The difference in adopting a 3T scanner is related to the better spatial resolution that is possible to acquire with this scanner technology. Moreover, high-resolution phased array coils (with 8–16 channels) are necessary to optimize the image quality. Currently, it is suggested to not perform an exam for endometriosis detection with a low magnetic field or open MRI because of the resulting sub-optimal image quality [66].

From a technical point of view, the use of a high-resolution acquisition sequence (3 mm or less) in the different spatial planes (axial, sagittal and coronal) in TSE-T2w are fundamental to evaluating DE. The use of the TSE T1w and the TSE T1W fat sat is requested to detect the presence of blood products in the adnexa (for the detection of endometrioma) [66].

Moreover, the spatial orientation (with the use of oblique axial) of the acquisition planes is very important because it allows for detecting and exploring more challenging areas, such as the USL, as demonstrated by Bazot et al. [67].

In the MRI assessment of endometriosis, the use of contrast material is debated because in most cases, it does not add any significant information; however, it is suggested that the adoption of CM could help with distinguishing DE from other pelvic inflammatory conditions, in particular, it is useful to differentiate a ruptured endometrial cyst from a tubo-ovarian abscess [68].

### 3.1. Ovarian Endometriosis

Ovarian endometriosis can contain varying different types of tissues, including proteins and fluids, but its pathognomonic characteristic is related to the presence of blood products at different stages of catabolism [69]. Usually, typical endometriomas have a very high signal in T1W and t1W FAT SAT and a low signal in T2W. Due to the presence of hemoglobin products at different stages in the endometriomas, it is possible to also detect the so-called “shading effect” in the T2W that shows the hematocrit level within the endometrioma.

In the MR assessment of an endometrioma, another potential diagnosis issue is differentiating endometrioma from a luteal hemorrhagic cyst because both lesions contain blood. In this case, differentiating the stage of the hemorrhage (acute versus chronic) could be useful and this could be obtained in the case of the presence of the T2 dark spot sign that is associated with the presence of chronic hemorrhage and, therefore, can be very helpful in differentiating endometriomas (positive for the T2 dark spot sign) from a luteal hemorrhagic cyst (negative for the T2 dark spot sign due to the acute hemorrhage phase) (Figure 6). There are currently no studies in the literature that evaluated the diagnostic accuracy in this anatomical location.

### 3.2. Superficial Endometriosis

Small superficial peritoneal implants are defined as those lesions that are <1 cm in diameter; these are the first grossly recognizable lesions on the surface of pelvic organs or the pelvic peritoneum [70]. Nodules of superficial endometriosis are challenging to detect with MRI, in particular, those lesions with a size of a few millimeters, but in those cases with the presence of blood products within it, it is possible to detect these nodules due to the high signal in T1 SPIR [70]. Sometimes, these lesions can be detected as a cluster of lesions (cystic or nodular) with a hyperintense signal in T1W with variable T2W appearance [70]. There are currently no studies in the literature that evaluated the diagnostic accuracy in this anatomical location.

### 3.3. Deep Endometriosis

In the case of nodules of endometriosis involving the bladder, it is possible to find two different patterns: localized involvement with evidence of nodularity in the bladder wall or/and diffuse wall thickening [71]. The characteristic low T2W signal can be associated with an intermediate signal in T1W with or without spots of high signal intensity on T1-weighted images, which represents blood products [72]. After the administration of contrast material in some cases, it is possible to observe a modest enhancement. According to the literature, in the diagnosis of bladder endometriosis, the sensitivity of MR is 88% with a specificity of up to 99% and a diagnostic accuracy of about 98% [73].

The MRI study of the ureter can be difficult [74] and MRI can show direct and indirect signs. The direct signs of extrinsic ureteral involvement are the detection of a nodule strictly adjacent to the ureter, which is characterized by a low signal on T2W due to the presence of fibrosis. In the case of the presence of blood products, high signal (T1W and T2W) foci can be detected. The indirect signs are the following: retractile periureteral adhesions (visible as low-signal-intensity lines with angular deviation) or ureteral dilatation cranial to the obstruction site [74]. A more infrequent occurrence is hydronephrosis, which is easily detectable with T2W (sensitive to the water) or with the more advanced T2W sequences and delayed contrast-enhanced 3D with higher spatial resolution [32] and magnetic resonance urography. For the detection of endometriosis in the ureter, Sillou et al. [75] demonstrated that MRI is more sensitive than surgery (91% vs. 82%) but less specific (59% vs. 67%)

A widely studied target with MRI in the endometriosis involvement is the USL (Figure 7); this is considered a complex target due to its small size and, as indicated by Bazot et al., the use of optimized high-resolution T2W oblique axial images can help with raising the sensitivity in detecting these lesions. As a general rule, the USL nodules of endometriosis should be suspected in the case of nodularity but also in the case of a thickened USL. The signal feature is a low signal because it is uncommon to find nodules with hemorrhagic components in the USL [66]; if this condition occurs, hyperintense spots may be observed in T2W, indicating dilated ectopic endometrial glands that can have either high or low T1W signal intensity according to the presence or absence of hemoglobin products.

As a topographical analysis, the most frequently involved area is the proximal medial portion of the USL. In a recently published meta-analysis, Medeiros et al. [76] found a sensitivity of 85% and a specificity of 80% for the MRI detection of USL endometriosis.

Furthermore, in the case of vaginal involvement, it is possible to identify direct and indirect findings. In the case of a direct finding, a nodule (with a low T2W signal) or the thickening of the parietal wall can be detected; in this case, the signal of the T1W sequence is also related to the presence and amount of hemoglobin in the hemorrhagic foci [77]. As an indirect sign, it is possible to find the obliteration of the pouch of Douglas that could be a sign of posterior vaginal involvement. In case of vaginal wall involvement, MRI could guide the surgery since the removal of the upper part of the vaginal wall is still mandatory for eradication [78].

The final localization of the DIE is the rectosigmoid wall, which is characterized by four layers, namely, serosa, outer longitudinal muscularis, inner circular muscularis and mucosa [79]. Endometriosis is an extra-rectal wall pathology and, therefore, the layers’ involvement moves from the serosa to the mucosa in the aforementioned order. Therefore in most cases, the serosal surface is the only layer involved with the nodular implants. As previously described, the signal characteristic is the same as the other localizations, with a low signal in T2W (isointense compared to muscle) [70]. The overlying layer, which is hyperintense on T2-weighted images, at the luminal side of the bowel wall corresponds to (sub)mucosal thickening, as determined by non-specific inflammation [70] (Figure 8 and Figure 9). With the advanced MRI techniques and, in particular, with the use of the 3T high-field technology, it is possible to detect the presence of rectal wall involvement and, more importantly, to characterize the degree of the layers’ involvement (from the serosa to the mucosa). A study from Abrao et al. [80] showed a sensitivity and specificity of 83% and 98%, respectively. Saba et al. [81] showed that MRI and tenderness-guided TVUS have similar sensitivities and specificities (73% and 90% for MRI and 73% and 86% for tenderness-guided TVUS, respectively).

Regarding parametrial involvement, MRI imaging of retraction of the obliterated umbilical artery toward the abdominal cavity was shown to be associated with the presence of endometriosis in the ipsilateral paracervix [82]. There are currently no studies in the literature that evaluated the diagnostic accuracy of MRI in this anatomical location.

In a meta-analysis, Guerriero et al. [83] compared the accuracy of TVUS and MRI in diagnosing DE. According to the results of this meta-analysis, the diagnostic performances of TVUS and MRI are similar for detecting DE involving the rectosigmoid, USLs and rectovaginal septum. A recent meta-analysis reported similar results regarding rectosigmoid DE [84].

However, for the USL, rectovaginal septum and vaginal DE, a new meta-analysis reported that MRI shows better sensitivity than transvaginal ultrasound [85].

Among the recent advances in imaging, the MRI/TVUS fusion technique is a new non-invasive diagnostic tool that could be useful in clinical practice because it may allow for improving the TVUS detection of some abnormalities using MRI and vice versa [86]. There is no consensus on the optimal requirements because of the limited number of published works about this technique [86]. In obstetrics and gynecology, preliminary studies suggested the potential value of fusion imaging techniques in diseases, such as DE, as well as scars from cesarean sections and gynecological cancers [86]. Fusion imaging allows for the superimposition of the MRI and US images, providing real-time imaging capabilities and high tissue contrast [87]. However, the disadvantage is that two exams need to be performed instead of one. Initial results show promising accuracy and combine the positive effects of both MRI and TVUS imaging [87]. However, before implementing this technique in daily practice, improvement in the software is necessary [88].

## 4. Other Imaging Techniques

Although not a substitute for the methodologies described above, some additional radiological techniques were proposed for some locations and specific purposes. We reviewed the most common techniques used in some locations of endometriosis, in particular, for the rectosigmoid colon. These techniques are the following: double-contrast barium enema, rectal endoscopic ultrasonography, multidetector computed tomography enema and computed tomography colonography (also known as a virtual colonoscopy).

The double-contrast barium enema (DCBE) was historically the first exam used to diagnose bowel endometriosis [89] but is currently not used due to the evolution of CT-based approaches. This technique involves injecting barium, followed by air, into the rectosigmoid colon. Bowel preparation is required before the examination. Barium is instilled while the patient lays in the left-side-down lateral position and, after intestinal hypotonization is performed, room air is then insufflated into the colon [89,90,91,92,93,94]. The effective dose is less than 5 mSv [94]. The whole procedure lasts an average of 7–15 min [91,92,93,94]. The examination is usually well tolerated, and approximately 10–20% of the patients report mild discomfort [88,90].

The limit of this approach is due to the fact that it is a pure luminographic approach and, therefore, endometriosis is suspected when the bowel lumen is narrowed at any level from the sigmoid to the anus [89,90]. The main advantage of DCBE is that it provides a complete overview of the entire colon via retrograde distention, for example, endometriotic cecal lesions could be diagnosed [91]. DCBE is easy to perform, is easily reproducible and it has a low cost [91]. DCBE is also less expensive than MRI [95,96,97]. As with many imaging modalities, its accuracy is operator-dependent [90,91].

DCBE offers indirect information because the cause of the lumen narrowing is not visible and requires the use of more advanced cross-sectional imaging that is nowadays becoming the first-line standard in this scenario [92]. Incomplete bowel distention, residual feces and incomplete passage of the contrast medium are additional factors that can hamper the diagnosis using DBCE [90,91]. Additionally, this technique cannot assess other anatomical areas that ultrasound and MRI can assess.

Another technique described by some authors is rectal endoscopic ultrasonography (REU). REU was traditionally used to stage rectal cancer [98], but in the last 20 years, it has also been used for the diagnosis of bowel endometriosis [99,100,101,102]. A rigid liner probe and a flexible endoscope with a lateral view and a 7.5–12 MHz convex echo probe equipment are used [99,102,103]. REU, like TVUS, allows for distinguishing the layers of the bowel wall, and the lesions appear as rounded or triangular hypoechogenic lesions infiltrating and thickening the intestinal wall [102,103]. The exam is usually well tolerated in approximately 80% of the patients [98]. The major limitation of REU is the possibility of evaluating only the distal part of the bowel (rectosigmoid) [104] and, for this reason, TVS or MRI should always be used to evaluate the presence of other lesions. A recent meta-analysis that compared the performances of TVUS, REU and MRI in the diagnosis of DE found that REU has a similar accuracy in comparison with the two other techniques but is more invasive [105].

Another imaging technique that can be used for the detection of endometriosis is the multidetector computerized tomography enema (MDCT-e). This is an MDCT with colonic distension obtained with water. This technique is widely used for the imaging detection and characterization of colorectal cancer [106], but due to the relevant information related to intestinal wall involvement in recent years, this technique was also adopted for the diagnosis of bowel endometriosis. As a general rule, preparation in the form of bowel cleansing is requested before performing this exam. Colonic distention is performed before beginning the exam by injecting warm water into the anal canal with the previous administration of an intramuscular injection of hyoscine-N-butyl bromide to obtain intestinal hypotonization [107,108]. Then, the administration of contrast material is performed, followed by a CT acquisition in the venous phase. The overall procedural time is usually less than 30 min. MDCT-e is usually well tolerated by patients [107,108] and offers excellent imaging visualization of the bowel wall and the typical appearance of the endometriosis involvement is the presence of a nodule with contrast enhancement infiltrating the colon wall. When there is no evidence of a fat plane between the bowel wall and the nodule, the infiltration of the intestinal muscolaris propria should be considered [109,110]. One of the advantages of MDCT-e is the potential to study the entire bowel with a differential diagnosis from other pathologies, such as cancer or inflammatory disease [106]. Nisenblat et al. [10] found that MDCT-e had the highest diagnostic performance for the rectosigmoid compared with other imaging techniques (TVUS, REU, MRI). Due to the radiation necessary to perform this type of exam, with an average radiation dose between 12 and 15.8 mSv [106], it is important to carefully verify if and when it is necessary to perform it due to the average age of female subjects that are at high risk of radiation-related damage. Another limitation is related to the administration of iodinated CM, which can cause adverse reactions [111]. Due to these disadvantages, MDCT-e is currently not routinely used as a first-line investigation for patients with suspicion of bowel endometriosis [109].

A different approach is computed tomography colonography (CTC), also named a virtual colonoscopy. This technique was initially developed for the study and characterization of polyps and colorectal cancers [112] but was subsequently adopted for the diagnosis of bowel endometriosis. For this CT procedure, pre-procedural cleansing is required but no need for water administration through a rectal catheter is necessary because to obtain distension of the colon, air or CO_2_ is pumped in via a flexible tube inserted into the rectum [107,113,114,115,116,117] with the patient in the left lateral decubitus and spasmolytic agents may be administered to facilitate the distention by relieving colonic spasms [116,118,119].

The contrast agent is administered orally at each meal on the day before CTC and it can be both an iodine-based solution or a barium-based solution [113,114,115]. There is no consensus on the most effective method of fecal tagging [116].

Furthermore, in CTC, venous administration of contrast material is necessary if the diagnosis of endometriosis is suspected when there is evidence of a nodule, which can be visible protruding on the profile of the distended colon (direct sign) or when tethering, or when there is flattening of the bowel wall and retraction of the mucosa (indirect sign) [120]. CTC is usually minimally painful and usually takes less time compared with MDCT-e, with examinations ranging between 15 and 24 min long [113]. An important advantage is the extraparietal information that can be gained [114]. The limitations are the same as a CT; these are related to the administration of iodinated contrast material and the risk related to the radiation dose (the average value is from 5 to 10 mSv).

It is important to underline that CTC should not be considered as an alternative to TVUS or MRI because these provide a better assessment of deep pelvic endometriosis, ovarian endometriomas and uterine adenomyosis [118].

A study conducted in 2020 by Biscaldi et al. [113] showed that CTC and magnetic resonance rectal enema (MR-e) are reliable techniques for determining the presence and extent of rectosigmoid endometriosis and that they are characterized by similar diagnostic performances.

Although performed in small study populations and mainly in the case of extragenital endometriosis, positron emission tomography–computed tomography with 16α-[18F]fluoro-17β-estradiol (FDG PET/CT) was proposed as a feasible technique [121,122]. In addition, Balogova et al. stated that, in some cases, the presence of endometriosis interferes with the FDG PET/CT performed for another indication [123].

Table 2 summarizes the advantages and disadvantages of the techniques described in this paper, apart from FDG PET/CT due to the very preliminary studies present in the literature.

To conclude, with advances in imaging techniques, much has already been done in recent years for the improvement in the diagnosis of endometriosis, both with an increase of the sensitivity and specificity of those techniques and preferentially using a less invasive diagnostic approach (relative to surgery). Certainly, more studies are needed to further expand knowledge in this area.

## Figures and Tables

**Figure 1 diagnostics-12-02960-f001:**
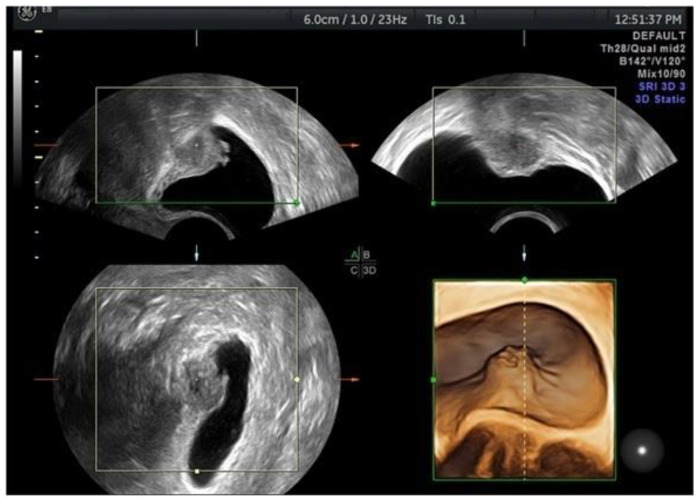
TVUS imaging method: a bladder endometriotic nodule with 2D TVUS (black and white images) and with 3D TVUS (color image).

**Figure 2 diagnostics-12-02960-f002:**
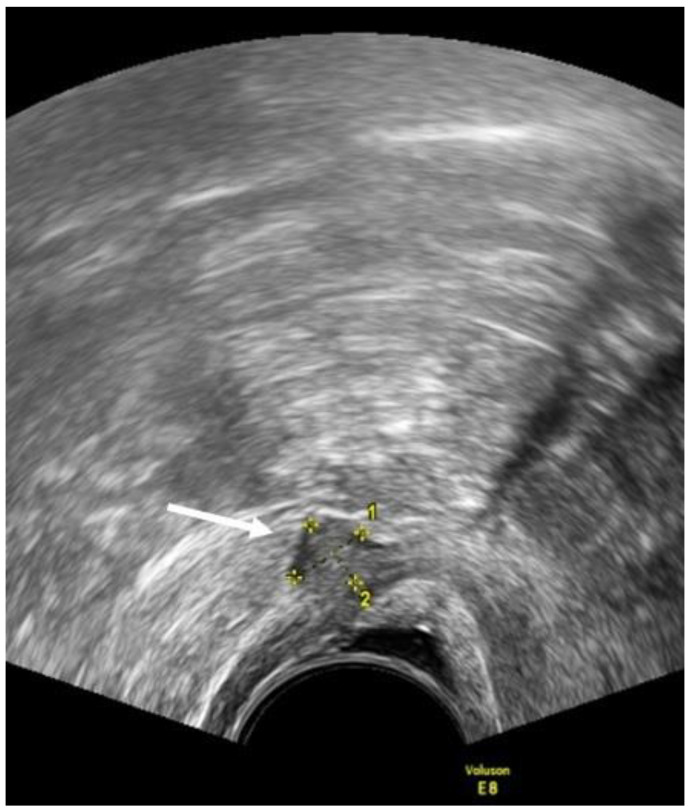
TVUS imaging method: the arrow and calipers show a nodule of the rectovaginal space in the sagittal plane.

**Figure 3 diagnostics-12-02960-f003:**
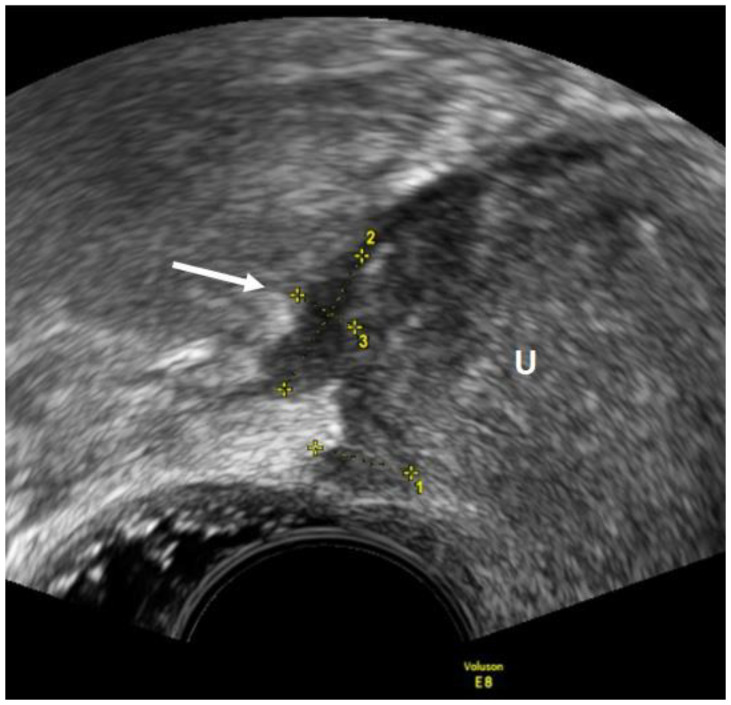
TVUS imaging method: USL lesion indicated by an arrow and calipers in the sagittal plane. U: uterus.

**Figure 4 diagnostics-12-02960-f004:**
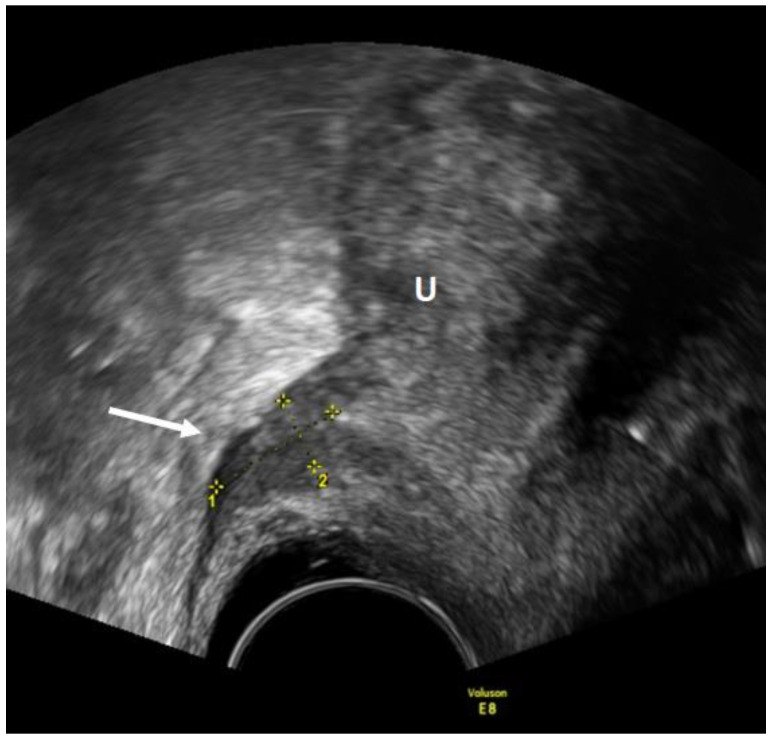
TVUS imaging method: the arrow and the calipers show a posterior vaginal fornix nodule in the sagittal plane. U: uterus.

**Figure 5 diagnostics-12-02960-f005:**
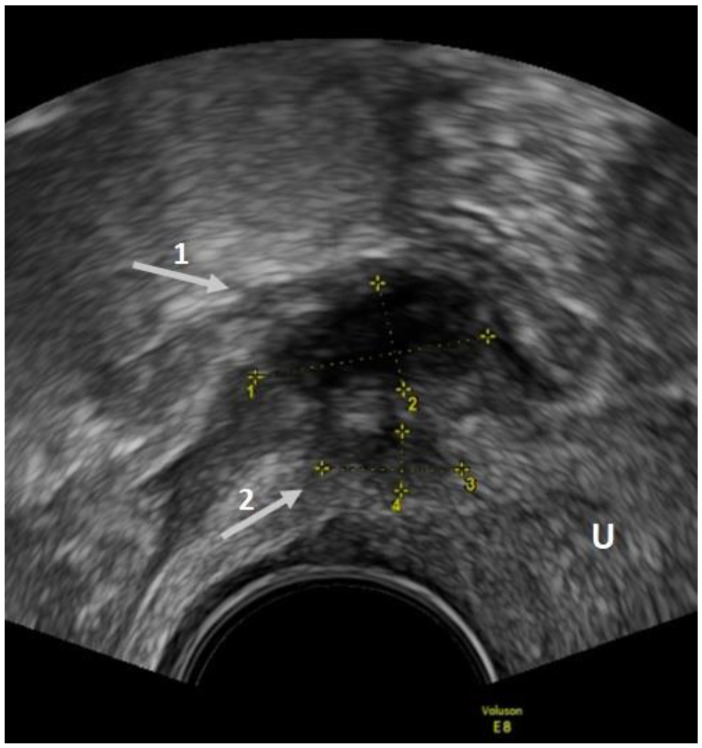
TVUS imaging method: arrow 1 shows a rectosigmoid nodule, while arrow 2 shows a USL nodule in the sagittal plane.

**Figure 6 diagnostics-12-02960-f006:**
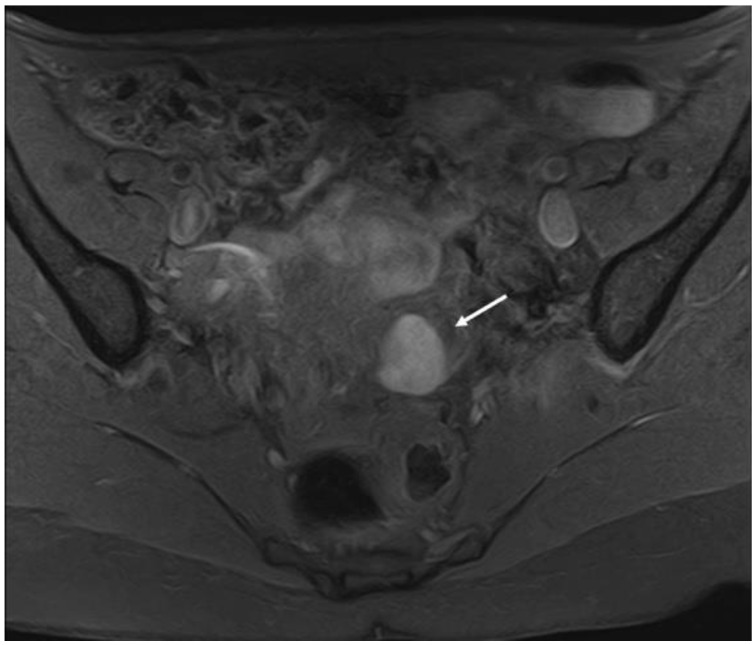
MRI imaging method: an endometrioma indicated by an arrow in the axial plane.

**Figure 7 diagnostics-12-02960-f007:**
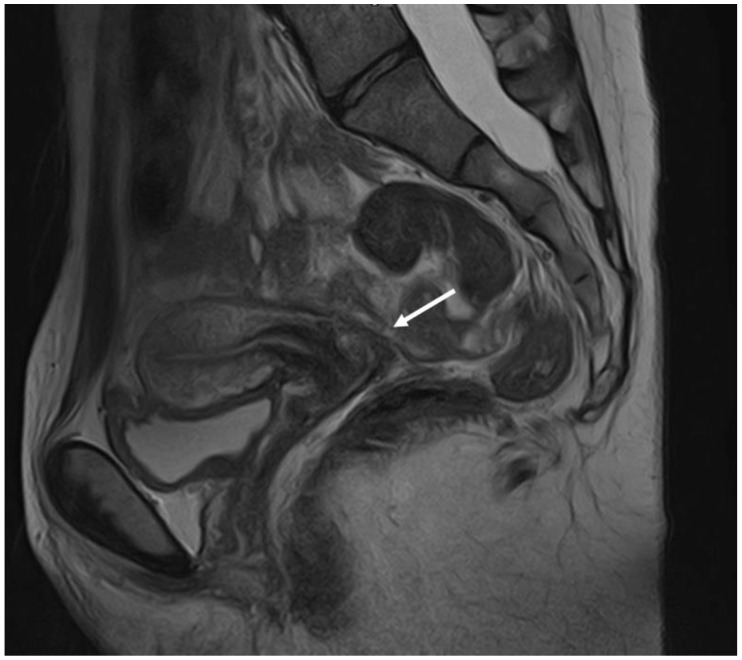
MRI imaging method: a retrocervical nodule indicated by an arrow in the sagittal plane.

**Figure 8 diagnostics-12-02960-f008:**
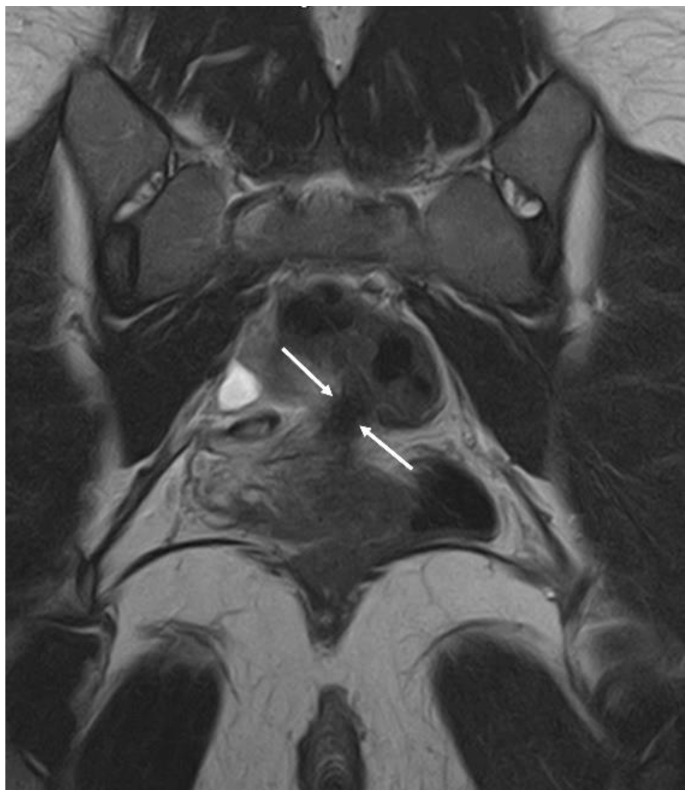
MRI imaging method: a nodule of rectosigmoid colon between the two arrows in the coronal plane.

**Figure 9 diagnostics-12-02960-f009:**
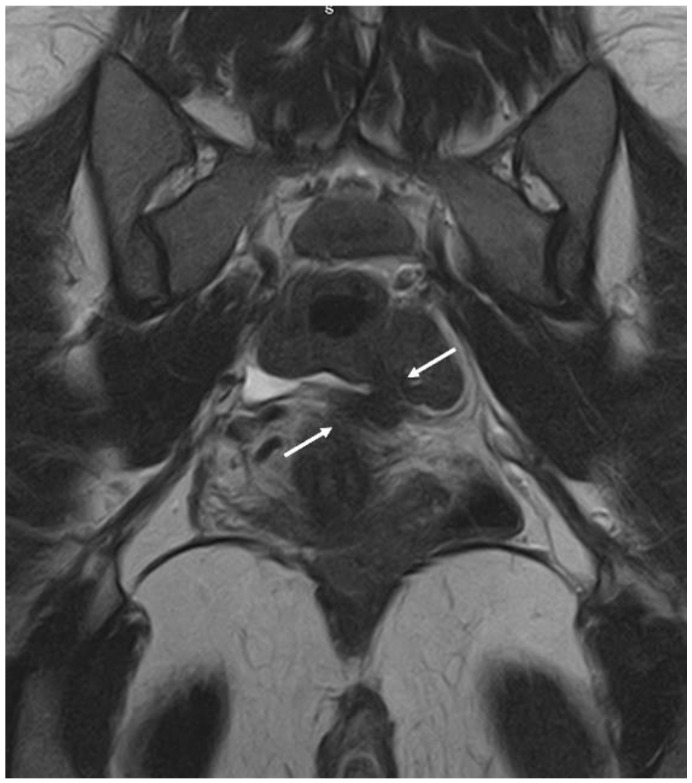
MRI imaging method: a nodule of the USL between the two arrows in the coronal plane.

**Table 1 diagnostics-12-02960-t001:** The different and more recent endometriosis classification/staging system based on ultrasound.

Classification/Staging System	Pub. Year	Based on	Based Only on Ultrasound	Aim of Classification
#Enzian classification [19]	2021	Surgical observation/MRI/ultrasound	No	Description of endometriosis
Endometriosis Fertility Index (EFI) [47]	2021	Surgical observation/ultrasound/clinical examination	No	Probability of pregnancy after endometriosis surgery
Adhesion scoring system [48]	2020	Ultrasound	Yes	Prediction of the pelvic adhesions
ENDORECT * [49]	2019	MRI/ultrasound/clinical examination	No	Prediction of rectosigmoid involvement
Preoperative ultrasound-based endometriosis staging system (UBESS) [50,51,52]	2016	Ultrasound	Yes	Prediction of the level of complexity of laparoscopic surgery

* Based on UBESS 3.

**Table 2 diagnostics-12-02960-t002:** The advantages and disadvantages of the different imaging techniques used for the diagnosis of endometriosis.

Imaging Technique	Time	Advantages	Disadvantages	Sensitivity	Specificity	Rad. Dose
TVUS	15–20 min	High specificity and high sensitivity for ovarian endometriosis High accuracy in detecting deep endometriosis and POD obliteration High tolerability Cost-effective Offers the opportunity to provide visual evidence to people Dynamic nature for organ mobility Consensus about the descriptions regarding the different locations Allows for anatomical mapping	Limited ability to detect superficial endometriosis The detection of deep endometriosis requires highly trained sonographers Operator dependent Examination may be considered painful	Depends on the location: - Bladder 62% - Rectosigma 95% - ULS 53% - Vaginal fornix 58% - Rectovaginal septum 49% - Parametrium 31%	Depends on the location: - Bladder 100% - Rectosigma 97% - USL 93% - Vaginal fornix 96% - Rectovaginal septum 98% - Parametrium 98%	None
MRI	30–40 min	Images obtained appear the same to all viewers Overall high accuracy in detecting DE and extra-pelvic endometriosis Allows for anatomical mapping Offers an opportunity to provide visual evidence to people Not painful No radiation exposure	Operator-dependent Static assessment Variable imaging protocols reported in the literature Less accurate in defining the bowel depth of invasion Limited ability to detect superficial endometriosis No consensus on how to describe the findings Higher cost compared with ultrasound Training specifically in endometriosis diagnosis needed	Depends on the location: - Bladder 88% - Rectosigma 73% - ULS 85%	Depends on the location: - Bladder 99% - Rectosigma 90% - USL 80%	None
DBCE	7–15 min	Complete overview of the entire colon via retrograde distention Less expensive than REU and MRI	It does not allow for identifying the cause of the mass effect Low specificity Only used in rectosigmoid endometriosis	42.9–100% *	93–100% *	<5 mSv
REU	15–20 min	Useful in a virgo patient Estimates the distance between lesions and the anal verge	Investigating only the distal part of the bowel (rectosigmoid) Poor sensitivity for ovarian endometrioma It does not allow for identifying anterior pelvis lesions Only used in rectosigmoid endometriosis	88.9–97.1% *	89.4–93.1% *	None
MDCT-e	30 min	Accurate and reproducible in diagnosing intestinal endometriosis Assessing endometriosis’ characteristics: the largest diameter of the nodule, the distance between the distal part of the nodule and the anal verge, and ddepth of infiltration of endometriosis in the intestinal wall	Administration of iodinated contrast medium and radiation exposure Only used in rectosigmoid endometriosis	93.3–100% *	96.6–100.0% *	12–15.8 mSv
CTC	15–24 min	High spatial resolution It allows for estimating the degree of intestinal stenosis Minimally painful	Radiation exposure and the process may require the administration of an iodinated contrast medium Only used in rectosigmoid endometriosis	68–96% *	48–86.7% *	9 mSv

DCBE: double-contrast barium enema; REU: rectal endoscopic ultrasonography; MDCT-e: multidetector computed tomography enema; CTC: computed tomography colonography. * Considering only studies with more than 50 patients.

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
