# Peer review of "Advances in Imaging for Assessing Pelvic Endometriosis"

_diagnostics, 2022, doi:10.3390/diagnostics12122960_

Round 1
Reviewer 1 Report
I really appreciate this useful and well-written manuscript. The submitted review paper is comprehensively and informative. The pictures are excellent and educational. The subject is of the greatest clinical importance. Here my remarks: 1) The “Authors” line contains an inappropriate sequence of first and last names: please see how they are converted to the bibliographic data ("Stefano, G.; Silvia, A.; Mariachiara, P" etc. instead of "Guerriero S, AjossaS, Pagliuca M" etc.) 2) The role and limitations of the PET-CT as a diagnostic tool for endometriosis are missing and this omission should be corrected. Please see and discuss: a) Cosma S et al. Accuracy of a new diagnostic tool in deep infiltrating endometriosis: Positron emission tomography-computed tomography with 16α-[18F]fluoro-17β-estradiol. PMID: 27558211. b) Balogova S et al. Interference of Known or Suspected Endometriosis in Reporting FDG PET/CT Performed in Another Indication. PMID: 35119396 c) Mulette P et al. Pulmonary cavitations with increased 18F-FDG uptake revealing a thoracic endometriosis: A case report. PMID: 34678890 3) The selection of the analyzed works is convincing, the high number of self-citations (18) is justified by their high quality. However, this review omits several papers that made significant contributions to endometriosis imaging (sometimes with groundbreaking findings). I strongly recommend to discuss the following studies dedicated to the imaging of, e.g., bowel endometriosis, Douglas endometriosis, abdominal endometriosis, tubal endometriosis, ureteral endometriosis etc: a) Savelli L et al. Endometriosis of the abdominal wall: ultrasonographic and Doppler characteristics. PMID: 21793086. b) Puppo A et al. Intraoperative Ultrasound for Bowel Deep Infiltrating Endometriosis: A Preliminary Report. PMID: 32991006. c) Stepniewska AK et al. Role of ultrasonographic parameters for predicting tubal involvement in infertile patients affected by endometriosis: A retrospective cohort study. PMID: 34418594. d) Stepniewska AK et al. Comparison of Virtual Ultrasonographic Hysteroscopy with Conventional Hysteroscopy in the Workup of Patients Who Are Infertile. PMID: 32197993. e) Leonardi M et al. Diagnostic accuracy of transvaginal ultrasound for detection of endometriosis using International Deep Endometriosis Analysis (IDEA) approach: prospective international pilot study. PMID: 35561121.Author Response
Reviewer 1
I really appreciate this useful and well-written manuscript. The submitted review paper is comprehensively and informative. The pictures are excellent and educational. The subject is of the greatest clinical importance.
Here my remarks:
1) The “Authors” line contains an inappropriate sequence of first and last names: please see how they are converted to the bibliographic data ("Stefano, G.; Silvia, A.; Mariachiara, P" etc. instead of "Guerriero S, AjossaS, Pagliuca M" etc.)
We modified the text according with the reviewer’s suggestions
2) The role and limitations of the PET-CT as a diagnostic tool for endometriosis are missing and this omission should be corrected. Please see and discuss: a) Cosma S et al. Accuracy of a new diagnostic tool in deep infiltrating endometriosis: Positron emission tomography-computed tomography with 16α-[18F]fluoro-17β-estradiol. PMID: 27558211. b) Balogova S et al. Interference of Known or Suspected Endometriosis in Reporting FDG PET/CT Performed in Another Indication. PMID: 35119396 c) Mulette P et al. Pulmonary cavitations with increased 18F-FDG uptake revealing a thoracic endometriosis: A case report. PMID: 34678890
We added these references to the revised manuscript. Also some sentences have added to the revised manuscript
3) The selection of the analyzed works is convincing, the high number of self-citations (18) is justified by their high quality. However, this review omits several papers that made significant contributions to endometriosis imaging (sometimes with groundbreaking findings). I strongly recommend to discuss the following studies dedicated to the imaging of, e.g., bowel endometriosis, Douglas endometriosis, abdominal endometriosis, tubal endometriosis, ureteral endometriosis etc: a) Savelli L et al. Endometriosis of the abdominal wall: ultrasonographic and Doppler characteristics. PMID: 21793086. b) Puppo A et al. Intraoperative Ultrasound for Bowel Deep Infiltrating Endometriosis: A Preliminary Report. PMID: 32991006. c) Stepniewska AK et al. Role of ultrasonographic parameters for predicting tubal involvement in infertile patients affected by endometriosis: A retrospective cohort study. PMID: 34418594. d) Stepniewska AK et al. Comparison of Virtual Ultrasonographic Hysteroscopy with Conventional Hysteroscopy in the Workup of Patients Who Are Infertile. PMID: 32197993. e) Leonardi M et al. Diagnostic accuracy of transvaginal ultrasound for detection of endometriosis using International Deep Endometriosis Analysis (IDEA) approach: prospective international pilot study. PMID: 35561121.
According to the reviewer’s suggestions some sentences have been added in the revised version about tubal involvement and intraoperative ultrasound. Also some comments about the paper of Leonardi have been added in the revised version. On the contrary the paper of Savelli has not been included because related to abdominal wall endometriosis and reviewer 2 asked to include in the review only pelvic endometriosis. The other paper of Stepniewska is about hysteroscopy which is not the topic of the present review.
Reviewer 2 Report
Dear Authors,
This article focuses on an important topic. I have carefully revised the manuscript entitled “Advantages in Imaging for Assessing Endometriosis” by Guerriero Stefano and colleagues. Despite the interesting aim, the manuscript has several flaws that require corrections to make it suitable for publication in the journal. The particularities and novelty of the article are not very well underlined in the results and conclusions sections. Given the bibliography, it is clear that the authors made a complete review of the literature beforehand.
However, some suggestions could improve the quality of the article:
- In the title, you should specify pelvic endometriosis because the entire article is written on US and MRI diagnosis of endometriosis with the pelvic location.
- Restructure the abstract
- Missing keywords
- The introduction must be rewritten – for example, nothing related to the epidemiology of the condition is discussed
- The article does not clearly specify whether it treats endometriosis in general or deep endometriosis "In this review, we describe the techniques used for diagnosis of DE."
- In section 2.1, it should be mentioned that there are ultrasound features related to age and that endometriosis mainly affects women of childbearing age, and pre- and postmenopausal changes are rarer and more difficult to diagnose.
- For a better understanding, DE mapping should be carried out in relation to the anatomical areas.
2.3.1 ...
2.3.2 ...
- The MRI/TVUS fusion technique should be exemplified by comparative TVUS and MRI images if possible.
Kind regards
Author Response
This article focuses on an important topic. I have carefully revised the manuscript entitled “Advantages in Imaging for Assessing Endometriosis” by Guerriero Stefano and colleagues. Despite the interesting aim, the manuscript has several flaws that require corrections to make it suitable for publication in the journal. The particularities and novelty of the article are not very well underlined in the results and conclusions sections. Given the bibliography, it is clear that the authors made a complete review of the literature beforehand.
However, some suggestions could improve the quality of the article:
- In the title, you should specify pelvic endometriosis because the entire article is written on US and MRI diagnosis of endometriosis with the pelvic location.
The term pelvic endometriosis has been added in the revised version as suggested by the reviewer
- Restructure the abstract
Thanks for the suggestion. The abstract has been completely rewritten
- Missing keywords
Some key words have been added.
- The introduction must be rewritten – for example, nothing related to the epidemiology of the condition is discussed
The introduction has been modified and partially rewritten adding some information about epidemiology but mainly focusing on the improvement of diagnostic accuracy (the main topic of this review) in the last years due to several reasons suggested in the text
- The article does not clearly specify whether it treats endometriosis in general or deep endometriosis "In this review, we describe the techniques used for diagnosis of DE."
Thanks of the suggestions. It was a typographical error that has been changed in endometriosis
- In section 2.1, it should be mentioned that there are ultrasound features related to age and that endometriosis mainly affects women of childbearing age, and pre- and postmenopausal changes are rarer and more difficult to diagnose.
Some sentences about the appearances of endometriomas during the different ages of the women have been added in the revised version
- For a better understanding, DE mapping should be carried out in relation to the anatomical areas.
2.3.1 ...
2.3.2 ...
Sub-headings added as suggested
- The MRI/TVUS fusion technique should be exemplified by comparative TVUS and MRI images if possible.
Unfortunately, at the moment we don’t have additional images of fusion to add.
Round 2
Reviewer 2 Report
I have reviewed this paper with particular interest because I work with problems caused by pelvic endometriosis, and I think it is important to emphasize the particularities related to the imaging evaluation. I think it is suitable for publication, and it will bode well for us to conduct investigations in this specific area and collect more data in the future.
Kind regards